# Sudden Death in Patients with a History of Kawasaki Disease under School Supervision

**DOI:** 10.3390/children9101593

**Published:** 2022-10-21

**Authors:** Mamoru Ayusawa, Hidemasa Namiki, Yuriko Abe, Rie Ichikawa, Ichiro Morioka

**Affiliations:** 1Faculty of Health and Medical Science, Kanagawa Institute of Technology, Kanagawa 243-0292, Japan; 2Department of Pediatrics and Child Health, Nihon University School of Medicine, Tokyo 173-8610, Japan; 3Division of Medical Education, Nihon University School of Medicine, Tokyo 173-8610, Japan; 4Department of Health Care Service Management, Nihon University School of Medicine, Tokyo 173-8610, Japan

**Keywords:** Kawasaki disease, coronary artery aneurysm, sudden death, school, exercise

## Abstract

We investigated the incidence of sudden death in students with a history of Kawasaki disease (KD) while under school supervision. Reports of sudden death in students with a history of KD during 1990–1999 and 2000–2009 were retrieved from the mutual aid system data. The student’s grade, sex, circumstances at the time of sudden death, final diagnosis, recommended restrictions on school activities, and intensity of physical activity at the time of sudden death were investigated. There were 11 cases from 1990 to 1999 and 3 from 2000 to 2009; KD was complicated with coronary artery aneurysm (CAA) in nine and one cases, respectively. The incidence of sudden death decreased by approximately 50% for KD history and 80% for KD with CAA between the two decades; however, the difference was not statistically significant. Of the 14 cases, 12 occurred during moderate-to-strenuous exercise; the restriction on exercise for students with KD complicated with CAA was not followed in at least five cases during 1990–1999, while three cases during 2000–2009 occurred without recommended restriction. Cases of sudden cardiac death decreased during 2000–2009, compared with those during 1990–1999. Special attention is required for students with a history of KD, particularly when complicated with CAA.

## 1. Introduction

Kawasaki disease (KD) is common in children under 4 years of age and is considered an entity of systemic vasculitis mainly affecting medium-sized arteries. After displaying characteristic clinical signs during the acute phase, coronary arteries may be involved, and coronary artery aneurysms (CAA) may develop. Small aneurysms tend to resolve spontaneously, but moderate and severe cases generally result in sequelae [1]. KD is currently the most common cause of acquired heart disease in countries and regions where antimicrobial agents are widely available, and rheumatic fever has been suppressed [2]. Dr. Tomisaku Kawasaki published the first case series in 1967 [3]. Since then, Japan has had the highest number of registered cases of KD in the world, with over 400,000 patients diagnosed from the 1960s to 2020. The number of patients with concomitant CAA is estimated to be ~25,000 [4]. Since the 1990s, when the efficacy of the high-dose, single infusion therapy of intravenous immunoglobulin (IVIG) was reported and became available for general use, the incidence and severity of CAA, and the prognosis of patients with KD have been improved. However, there is still concern about sudden death due to myocardial infarction caused by the thrombus occlusion of coronary aneurysms, although it occurs in a small percentage of patients [5].

Therefore, we conducted a study to determine the incidence of sudden death among school-aged children with a history of KD by reviewing the data from the mutual aid system for students.

## 2. Materials and Methods

### 2.1. Definition of the Database

The Japan Sport Council (JSC, Tokyo, Japan) administers the injury and accident mutual aid benefit system for students, which provides medical expenses and compensation for health problems caused by school accidents and disasters according to regulations upon submission of a report [6]. Over 97% of students nationwide are enrolled in the program, including elementary, junior high, and high school students, as well as students at special-needs schools and part-time schools. Almost all deaths are reported, and large compensations are paid.

We identified the cases of deaths reported to the JSC during the first decade (1990–1999) and the second decade (2000–2009) that were classified as sudden deaths, and extracted cases with a known history of KD and those confirmed on autopsy. The cases in which CAA was clearly documented in the report were described as having CAA. The cases of KD with a history of acute myocardial infarction (AMI), acute heart failure (AHF), and coronary artery bypass graft surgery (CABG) were also extracted from the records. From the initial pathological autopsy, the descriptions of the coronary arteries and myocardium were specifically extracted and recorded. The final diagnoses of AMI and AHF obtained from the autopsy were also extracted. Sudden death was defined by the JSC as “an endogenous death occurring during the first 24 h of illness, including cases resulting in death without recovery after longer than 24 h with life-support procedures.” “Under school supervision” is defined as “the period between the time a child leaves for school and the time the student returns home” and includes club activities and events under school supervision, even on holidays.

### 2.2. Investigated Items

The year of the sudden death and the student’s grade, sex, diagnosis, circumstances at the time of sudden death, final diagnosis including pathological autopsy, recommended restrictions on school activities, and intensity of physical activity at the time of sudden death were investigated in those cases in which KD was noted from the report to the JSC. The intensity of physical activity was classified into levels 1 to 5: level 1 was stationary; level 2 was daily activities without exertion, such as walking, eating, and studying; level 3 was daily activities with mild exertion, such as cleaning and carrying luggage; level 4 was performing exercises at a level that did not cause pain; and level 5 was sports games, competition, and strenuous training, and these levels were judged by the authors. In Japan, documentation is prepared and submitted by the attending physician indicating the need for restrictions in school activities for students with underlying medical conditions in order to manage the students’ health at school. The degree of exercise allowed for the medical condition is indicated and restricted by A, B, C, D, and E. A is indicated in the case necessary to stay home and to receive medical care or hospitalization, B allows only school attendance and prohibits all forms of exercise, C allows light exercise, D allows moderate exercise, and E allows strenuous exercise. Additionally, “prohibited” or “allowed” is written according to the allowance of the extracurricular club activities. For example, KD, E-allowed indicates that the student is exactly the same as other students and has no exercise restrictions but has a pre-existing condition such as KD. These details are written on a diagnostic form called School Activity Management Table, and the information is shared with school staff [7].

### 2.3. Statistical Comparison

For comparison between the 1st and 2nd decade, we calculated the proportion of KD cases among sudden deaths and sudden cardiac deaths and investigated if there was a statistically significant change by using the χ-square test.

## 3. Results

There were 969 sudden deaths during the first decade from 1990 to 1999, of which 703 were sudden cardiac deaths, and 11 cases (boys: 8, girls: 3) were reported as having a history of KD (Table 1). Among the 11 students who had a KD history, 6 students were confirmed as having coronary artery complications, including a case of KD with past CABG and a case of KD with a past history of AMI. The four cases of KD with CAA included one with bilateral CAA, another with right CAA, and two others with CAA of unknown laterality. The other five cases had a history of KD only, but the final diagnosis was AMI in three cases. Two patients with a final diagnosis of AHF were not confirmed as having CAA, AMI, or CABG (Table 1).

On the other hand, there were 526 sudden deaths in the second decade from 2000 to 2009, including 368 sudden cardiac deaths, and 3 patients were confirmed as having a history of KD. The data are presented in Table 2. All were boys. One case was diagnosed with KD and CAA, and the final diagnosis in this case was AMI. In the other two cases, CAA was not confirmed, and the final diagnosis in both cases was arrhythmia (Table 2).

Statistically, the number of cases described as having KD among sudden deaths or sudden cardiac deaths was compared between the first and second decades using the χ^2^ test to determine if there was a significant change.

The results showed that the proportion of sudden deaths in students with KD among all sudden deaths and the proportion of sudden deaths in those with KD among all sudden cardiac deaths decreased by approximately 50% in the second decade compared with those in the first decade, but it was not statistically significant based on the χ^2^ test. The frequency of sudden death among all sudden deaths and the frequency of sudden death among sudden cardiac deaths in patients with KD complicated with CAA decreased to about 20% in the second decade, compared with those in the first decade, but it was not statistically significant based on the χ^2^ test. (Table 3).

In terms of the circumstances at the time of sudden death, 9 of the 11 cases in the first decade occurred during level 4 or 5 exercises. In two of these cases, exercise restrictions on the school activity management table were not discovered among reports from their school, and the other cases needed no restriction or management. Of the six patients who died during level 4 or 5 exercises, only one was “E-allowed”, who died after a basketball game for 12 min, three were “D” who died while running approximately 18 km, 4 km, or after full participation to 40 min of a basketball game, and two were E-prohibited but collapsed and died while performing level 5 exercise 400 m or about 3 km. Two cases died while performing level 1 or 3 exercises, respectively. One of them was with a history of CABG surgery due to coronary artery complications and was managed as the “B” category by the school activity management table and died when sitting on a chair (level 1), and another was unknown about the recommended restriction on the school activity management table and died when he had begun toss batting (level 3), and his final diagnosis was AMI. The three cases in the second decade all occurred during level 5 exercise, and the school activity management table identified in two cases, in which one was described as having CAA and was E-allowed, and the other (case 12) was not certified as having CAA and written as ‘No restriction or management’ on his school activity management table. The estimated cause of death was arrhythmia in this case. Concerning the case 14, we could not discover the school activity management table. However, as we could find that he was not having CAA, probably no restriction or management were necessary. In cases 12 and 14, with no mention of CAA, and both collapsed after very strenuous sports activitiy. and their final diagnoses were “fatal arrhythmia” and “arrhythmia”.

## 4. Discussion

The coronary sequelae of KD are currently the most common acquired cardiac disease in developed countries and are feared because of the possibility of sudden death due to ischemic heart disease during the course of the disease. The onset of the disease is during childhood, primarily occurring in children below 4 years of age. However, after the acute phase, almost all patients are asymptomatic, whether with or without the presence of CAA. In such cases with sequelae, sudden death is a lifelong concern, but there are no reports that have comprehensively analyzed the actual situation in students.

Our study is limited to data from 1990 to 2009. Since the number of cases decreased to three in the second decade, it may be assumed that the number of cases will not increase thereafter because of further advances in the treatment to prevent coronary artery complications in the acute phase of KD. Japan probably experiences the largest number of cases of KD in the world, including those with sequelae. In this study, we used the data from the mutual aid benefit system as a means of determining the prevalence of sudden deaths in students with a history of KD. As noted in the Methods section, nearly all the students were enrolled in this program throughout the study period, and in nearly all cases of death, the detailed circumstances of the occurrence were reported.

Comparing each decade of the 1990s and 2000s, no statistically significant difference was obtained, but this might be due to the fact that sudden deaths were very rare in Japan, and the incidence was numerically small. However, a comparison of the incidences of sudden deaths in two decades showed a reduction of approximately 50% for all patients with KD and approximately 80% for those with CAA. Since this study is considered to be a near-total survey, it can be assumed that the number of sudden deaths from KD in schools has decreased.

During the first decade, 9 of the 11 cases had a coronary aneurysm, and some had already undergone CABG or had a history of AMI. Myocardial infarction was the cause of death in only one of the three sudden deaths during the second decade, while arrhythmia was the presumed cause in the other two cases. As this database was not hospital data, we could not confirm that arrhythmia was indeed the cause of death; however, since there was no mention of organic disease in the final certificates, arrhythmia seemed to be a possible cause of death. The severity or frequency of arrhythmia in patients with KD is unknown. However, QT prolongation and change in QT dispersion [8], and out-of-hospital cardiac arrest due to fatal arrhythmias have been reported after KD [9].

After the year 2000, anticoagulation with oral warfarin has been widely used in severe cases of CAA, and therefore, the incidence of myocardial infarction decreased [10]. This is a possible factor for the decrease in KD-related sudden death. However, a national surveillance report of the acute coronary syndrome in young adults from 2000 to 2009 reported at least 32 cases with a history of suspected KD with CAA among the eligible cases (mean age: 26 years), some with no outpatient management and some with self-interruption of medication use [11].

There is a limitation of this study. Although the history of Kawasaki disease with or without CAA is detected by the documents reported to the mutual aid system of JSC, we could not exactly detect whether those students were taking medicines or not, because the reports were written not by hospital workers but by school workers.

We are concerned about the possibility that patients with KD complicated with CAA who had been prevented from myocardial infarction by taking medication under parental supervision during school ages began to neglect their medication when they became adults, eventually leading to sudden death or out-of-hospital cardiac arrest. This may indicate that the age group at risk for KD-related sudden deaths is gradually shifting to young adults or older. On the other hand, we must encourage students or young adults to exercise adequately to prevent metabolic syndromes such as atherosclerosis or diabetes mellitus. It is difficult to determine what degree of intensity of sports activity or event is possible and adequate to engage in. We need to clarify the efficacy of the exercise stress ECG test, exercise or pharmacological stress echocardiography [12], and scintigram [13]. In Japan, since approximately at least ten thousand students have a history of Kawasaki disease every year, and 95% do not have coronary artery complications, it is not possible to follow all KD patients until they become adults.

In this sense, it is very important that KD patients with sequelae of CAA continue to undergo careful life-long outpatient management by attending physicians, and that they remain eligible for transitional care into young adulthood and beyond.

## Figures and Tables

**Table 1 children-09-01593-t001:** Cases of sudden death in students with a history of Kawasaki disease under school supervision (1991–1999).

Case	Year	Grade	Sex	Diagnosis	Situation at Onset	Final Diagnosis (by Autopsy, If Done)	Restriction *	Exercise Strength
1	1991	4	F	KD, p/o CABG	While sitting on a chair	None	B	1
2	1991	9	F	KD, s/p AMI	18 km road race; 40 m before the finish line.	Autopsy: LCAA Total occlusion, RCAA, Fibrosis at the posterior and lateral wall of LV. Dilative hypertrophy of RV, Weight of heart: 340 g.	D	5
3	1991	10	M	KD, CAA	1.7 km from the finish line of the 4.2 km endurance run	None	D	5
4	1993	9	F	KD, Bilateral CAA	After a full basketball game for 40 min	Autopsy was performed, but its result was unknown.	D	5
5	1993	11	M	KD, CAA	400 m running	AMI	E-prohibited	4
6	1994	11	M	KD	Fell down while riding bicycle to school.	AHF	N/D	4
7	1995	11	M	KD	2850 m running	AHF, AMI, MCLS	E-prohibited	5
8	1995	8	M	KD	After basketball practice	KD, AMI	N/D	4
9	1996	12	M	KD, RCAA	After a basketball game for 12 min	Autopsy: ischemic heart disease, atherosclerosis of coronary artery, RCAA (13 × 10 mm)	E-allowed	5
10	1998	7	M	KD	After tennis practice for 2 h	AHF	No restriction or management	4
11	1998	8	M	KD	After 2 swings of toss batting	AMI	N/D	3

Restriction *: The categories of restricion of school activity is defined and marked on the ‘School Acitivity Management Table’ by reffered doctors as follows; A: necessary to stay home and to receive medical care or hospitalization (no case in this case series), B: school attendance is alllowed and all forms of exercise are prohibited, C: light exercise is allowed(no case in this case series), D: moderate exercise is allowed, E: strenuous exercise is allowed. Additionally, “prohibited” or “allowed” is written according to whether the extracurricular club activities are allowed or prohibited. N/D: not discovered among reports from school to Japan Sports Council. Concerning Case 10, we could discover the corresponding direction written ‘No restriction or management’ on School Activity Management Table for this student.

**Table 2 children-09-01593-t002:** Cases of sudden death in students with a history of Kawasaki disease under school supervision (2000–2009).

Case	Year	Grader	Sex	Diagnosis	Circumstances	Final Diagnosis (by Autopsy, If Done)	Restriction *	Exercise Strength
12	2000	7	M	KD	After cycling for 14 km to school, walked 1.2 km, warmed up, and practiced. After playing in the first game of softball, practiced for 20 min and played in the second game without eating lunch. As a catcher, he collapsed and fell on his back while trying to throw a ball.	Arrhythmia (fatal)	No restriction or management	5
13	2004	8	M	KD, CAA	At the end of volleyball club activities (stretching, jogging 5 laps around the court, dashing 5 laps, then 2 laps between the line and the net), he felt lightheadedness and suddenly fell forward.	Myocardial infarction	E-allowed	5
14	2008	6	M	KD, No CAA	After two basketball games, he moved to the side of the court. A short time later, he began to act strangely and lost consciousness. An ambulance was called. No consciousness or pulse was detected. AED and CPR were performed immediately, and the patient was transported to the hospital by ambulance.	Arrhythmia, Cardiac arrest	N/D	5

Restriction *: The categories of restricion of school activity is defined and marked on the ‘School Acitivity Management Table’ by reffered doctors as follows; E-allowed: strenuous exercise and the extracurricular club activities are allowed. N/D: not discovered among reports from school to Japan Sports Council. Concerning Case 12, we could discover the corresponding direction written ‘No restriction or management’ on his school activity management table.

**Table 3 children-09-01593-t003:** Comparison of proportion of students with a history of Kawasaki disease among all sudden deaths or sudden cardiac deaths under school supervision.

Sudden Death with Past History of Kawasaki Disease	1st Decade (1990–1999)	2nd Decade (2000–2009)	Ratio of Incidence in 2nd Decade Compared to That in 1st Decade	χ^2^ Value	*p* Value
Proportion among total sudden deaths	11/969 (1.13%)	3/526 (0.57%)	50.2%	1.17	0.28
Proportion among total sudden cardiac death	11/703 (1.56%)	3/368 (0.82%)	52.1%	1.05	0.30
**Sudden Death with Coronary Artery Aneurysm Due to Kawasaki Disease**	**1st Decade** **(1990–1999)**	**2nd Decade (2000–2009)**	**Ratio of Incidence in the 2nd Decade** **Compared to That in 1st Decade**	**χ^2^ Value**	***p* Value**
Proportion among total sudden deaths	9/969 (0.93%)	1/526 (0.19%)	20.5%	2.80	0.094
Proportion among total sudden cardiac death	9/703 (1.28%)	1/368 (0.27%)	21.2%	2.66	0.10

## Data Availability

Some examples of the database are partly available for reference on the website of the Japan Sports Council; https://www.jpnsport.go.jp/anzen/anzen_school/anzen_school/tabid/822/Default.aspx (accessed on 18 September 2022).

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
