# Peer review of "Sudden Death in Patients with a History of Kawasaki Disease under School Supervision"

_children, 2022, doi:10.3390/children9101593_

Round 1

Reviewer 1 Report

Could the authors also comment regarding the usage of other drugs in treatment of Kawasaki disease at least for children included in the later group 

Also could the authors comment on the fact that most of the sudden death cases were after a sport effort. This could mean that we as pediatricians should not recommend to Kawasaki disease intense physical effort?

Author Response

(1)Could the authors also comment regarding the usage of other drugs in the treatment of Kawasaki disease, at least for children included in the latter group 

Thank you very much for Reviewer 1's comments and inquiries.

For the 1st inquiry, unfortunately, we can not know whether some medications are prescribed or not to each patient because the database in this study was analyzed from the reports written not by doctors but by the school teachers or caregivers. However, as for the latter group, the first and the third patients were not supposed to be treated by medication because there were no reports or health certificates written as having coronary artery complications, even though they were written as Kawasaki disease, and they died of other cause than myocardial ischemia including myocardial infarction.

We added the limitation about it with your pointing out below and the main text.; thank you.

There are some limitations of this study. At first, the history of Kawasaki disease with or without CAA is detected by documents reported to the Mutual Aid System of JSC. However, we could not precisely detect whether those students were taking medicines or not because reports were written not by hospital workers but by school workers.

(2) Also could the authors comment on the fact that most of the sudden death cases were after a sport effort. This could mean that we as pediatricians should not recommend to Kawasaki disease intense physical effort?

Thank you for another very important inquiry. We do not think that we need restrictions for students with a history of Kawasaki disease if the patients after Kawasaki disease do not have significant cardiac complications. Many studies report sudden deaths or out-of-hospital cardiac arrests (OHCA) occur in young people when they engage in strenuous or competitive sports. Such possibilities are very small but unexpected who will occur. So our cases who are not proven to have any cardiac complications can participate the regular sports activities. Unless they do, other problems, such as the progress of obesity and young arterial sclerosis.

We added the comment for this pointing out in Discussions.

On the other hand, we must encourage students or young adults to exercise adequately to prevent metabolic syndromes such as atherosclerosis or diabetes mellitus. It is difficult what degree of intensity of sports activity or event is possible and adequate to engage in. We need to clarify the efficacy of exercise stress ECG test, exercise or pharmacological stress echocardiography [12], and scintigram [13].

Reviewer 2 Report

Dear author,

This is an good update of consequences that could be seen in KD.

The results shows us that all children with KD should have regular check ups even in adult period.

Kind regards,

Dragana 

Author Response

Thank you very much for Reviewer 2's kind comment. 

If outside Japan, it may be better for the physicians to follow all the patients after Kawasaki disease considering the genetic factor of the atherosclerosis progression in the youth or dietary difference.

However, in Japan, numerous patients have a history of Kawasaki disease, and 95% do not have coronary artery complications. So it may be ideal that we can follow all the patients until they become 50-60' or older, but it seems that it is thought to be time and effort-consuming in our experiences. This problem does not have an exact answer until the present situation. 

We added this problem briefly in the discussion by your comment. Thank you.

Reviewer 3 Report

I reviewed your manuscript, and I suggest some minor changes according to the context and grammar. Please refer to the attached file. Red-lined parts are to be deleted, and yellow-colored parts are to be changed into the contents in memo boxes. Green-colored parts are to be checked again and need to be revised appropriately.

Thank you.

Author Response

I sincerely appreciate Reviewer 3 for the many precise proofreading of our manuscript. Most parts that were drawn with yellow (to change), green (to check and correct), and red (to delete) markers have been revised according to those suggestions.

However, the explanation of Tables 2 and 3 probably confused readers other than Japanese pediatric cardiologists.

To be understood precisely, we need to explain the categories ‘2B’, ‘3D’, ‘1D’, ‘2D’, ‘2E-prohibited’, and ‘3E-allowed’.

These categories are written on the school activity management table by the referral doctor caring for the concerned student.

The number ‘1’, ‘2’ or ‘3’ at the top of those categories mean ‘hospitalization necessary,’ ‘medical treatment is necessary,' or ‘observation in hospital is necessary,’ respectively.

The capital letters from A to D  show the degree of restriction written from line 82 to line 84, and additionally, ‘prohibited’ or ‘allowed’ is also shown after the capital letter concerning sports club activities.

However, with a change in the system of school heart screening and the management of the school activity of students, the number showing management in the hospital was thought to be unnecessary to show since around 2000. Therefore, we revise those parts by deleting numbers 1, 2, and 3 to smoothly understand and not confuse the number of the patient.

Concerning other expressions, N/D means we could not find any certificate or health check record from the reports submitted to the Mutual Aid System of Japan School Council. ‘No restriction or management’ means that the referral doctor submitted a school activity management table by writing as no restriction even though the student had a history of Kawasaki disease, probably because of no or mild coronary artery complication.

Those changes are reflected in the revised manuscript.